# The Association between Mechanical Power and Mortality in Patients with Pneumonia Using Pressure-Targeted Ventilation

**DOI:** 10.3390/diagnostics11101862

**Published:** 2021-10-10

**Authors:** Huang-Pin Wu, Chien-Ming Chu, Li-Pang Chuang, Shih-Wei Lin, Shaw-Woei Leu, Ko-Wei Chang, Li-Chung Chiu, Pi-Hua Liu, Kuo-Chin Kao

**Affiliations:** 1Division of Pulmonary, Critical Care and Sleep Medicine, Chang Gung Memorial Hospital, Keelung 20401, Taiwan; whanpyng@cgmh.org.tw (H.-P.W.); rocephen2000@yahoo.com.tw (C.-M.C.); 2College of Medicine, Chang Gung University, Taoyuan 33302, Taiwan; r5243@cgmh.org.tw (L.-P.C.); ec108146@cgmh.org.tw (S.-W.L.); swleu@cgmh.org.tw (S.-W.L.); b9302072@cgmh.org.tw (K.-W.C.); pomd54@cgmh.org.tw (L.-C.C.); 3Department of Thoracic Medicine, Chang Gung Memorial Hospital, Taoyuan 33302, Taiwan; 4Clinical Informatics and Medical Statistics Research Center, College of Medicine, Chang Gung University, Taoyuan 33302, Taiwan; phliu@mail.cgu.edu.tw

**Keywords:** mechanical power, driving pressure, pneumonia, acute respiratory distress syndrome, non-acute respiratory distress syndrome

## Abstract

Recent studies have reported that mechanical power (MP) is associated with increased mortality in patients with acute respiratory distress syndrome (ARDS). We aimed to investigate the association between 28-day mortality and MP in patients with severe pneumonia. In total, the data of 313 patients with severe pneumonia were used for analysis. Serial MP was calculated daily for either 21 days or until ventilator support was no longer required. Compared with the non-ARDS group, the ARDS group (106 patients) demonstrated lower age, a higher Acute Physiology and Chronic Health Evaluation II score, lower history of diabetes mellitus, elevated incidences of shock and jaundice, higher MP and driving pressure on Day 1, and more deaths within 28 days. Regression analysis revealed that MP was an independent factor associated with 28-day mortality (odds ratio, 1.048; 95% confidence interval, 1.020–1.077). MP was persistently high in non-survivors and low in survivors among the ARDS group, the non-ARDS group, and all patients. These findings indicate that MP is associated with the 28-day mortality in ventilated patients with severe pneumonia, both in the ARDS and non-ARDS groups. MP had a better predicted value for the 28-day mortality than the driving pressure.

## 1. Introduction

Ventilated patients with inappropriate ventilator settings may further develop lung injury. Studies have suggested several lung protective strategies to minimize ventilator-induced lung injury (VILI) [1,2,3]. These studies recommend specific mechanical ventilation settings for patients with or without acute respiratory distress syndrome (ARDS), including (1) low tidal volume ventilation (6 mL/kg predicted body weight (PBW)), (2) relatively higher positive end-expiratory pressure (PEEP), and (3) using an upper-limit goal for end-inspiratory plateau pressures of 30 cm H_2_O [1,3,4]. 

Amato’s study reported that driving pressure was most strongly associated with survival in patients with ARDS who received mechanical ventilation in different combinations of tidal volume and PEEP [5]. In patients with severe ARDS receiving extracorporeal membrane oxygenation (ECMO), higher driving pressure during the first three days of ECMO support was independently associated with increased mortality [6]. In patients with severe pneumonia without ARDS, higher driving pressure was associated with 28-day mortality [7]. Thus, driving pressure could be used as a non-invasive method of predicting lung injury in patients with and without ARDS. 

Recently, a new concept of safe mechanical ventilation using mechanical power (MP) was introduced [8]. The MP registered several underlying qualities of relevance. First, it imports static compliance, which induces VILI [9]. Second, it accounts for the final effect of PEEP. The PEEP is positively associated with VILI, but PEEP may decrease the lung-dependent VILI by reducing lung inhomogeneity and intratidal alveolar collapse–decollapse. Third, transpulmonary MP increases with the respiratory rate (RR) [10]. Serpa et al. reported that the MP of ventilation in the first 48 h is associated with mortality in critically ill patients in two observational cohorts. In patients with ARDS, initial MP was also associated with mortality [11,12,13]. These four studies all used a simplified equation proposed by Gattinoni et al. to calculate MP in patients with volume-controlled ventilation [8,14].

Therefore, we hypothesized that MP was a summary variable including all the components which can possibly cause VILI and had a better predicted value for mortality compared with driving pressure alone. We also tried to identify the association between MP and 28-day mortality in patients with pressure-controlled ventilation (PCV). 

## 2. Materials and Methods

### 2.1. Procedure

This study is a retrospective analysis of the database of our previous consecutively sampled observational cohort of patients with severe sepsis. This study was approved by the Institutional Review Board of Chang Gung Memorial Hospital, and the need for a written informed consent was waived (201700804B0 12 June 2017). Patients with severe pneumonia admitted to the medical intensive care unit (ICU) at the Chang Gung Memorial Hospital, Keelung from July 2007 to June 2010 were selected. The exclusion criteria included infections other than pneumonia, absence of invasive ventilator support, an unknown arterial partial pressure of oxygen (PaO_2_)/fraction of inspired oxygen (FiO_2_) ratio, or death on the day of admission. None of the included patients withdrew from this study. 

### 2.2. Disease Definitions

Due to the fact that the database was around 10 years old, disease definitions were renewed as follows: Pneumonia was defined as a new abnormal infiltration on the chest radiograph with respiratory symptoms or fever. Severe pneumonia was defined as pneumonia complicated by acute respiratory failure requiring intubation and mechanical ventilation with or without septic shock [15]. Sepsis and septic shock were defined according to the Sepsis-3 guidelines [16]. Sepsis was defined as a suspected or documented infection with an acute increase (≥2) in the Sequential Organ Failure Assessment points. Septic shock was defined as sepsis with a blood lactate level >18 mg/dL and hypotension that was unresponsive to fluid resuscitation, requiring vasopressors to maintain a mean arterial pressure ≥65 mm Hg during the first 3 days following ICU admission. Stage 2 or 3 acute kidney injury was defined according to the Kidney Disease Improving Global Guidelines (KDIGO) [17]. Disease severity was assessed with the Acute Physiology and Chronic Health Evaluation (APACHE) II score [18]. ARDS was defined according to the Berlin definition [19]. ARDS was evaluated via chest radiographs obtained after intubation with ventilator support. Patients who survived for at least 28 days from ICU admission were considered survivors.

### 2.3. Ventilator Settings in Our Previous Cohort

In our hospital, ventilated patients routinely received pressure-targeted ventilation. Volume-controlled ventilation (VCV) was not used due to the rapid change of peak pressure, which resulted in a risk of VILI. Following intubation, all patients with ARDS and non-ARDS were routinely administered a separate target tidal volume by PCV of approximately 6 and 10 mL/kg PBW, respectively. The goal was to maintain an inspiratory plateau pressure (Pplat) of less than 30 cm H_2_O. The PEEP level and FiO_2_ were adjusted to maintain a PaO_2_ greater than 60 mmHg or oxygen saturation by pulse oximetry (SpO_2_) greater than 90%. Ventilator settings were adjusted after 2 h of the first setting. Ventilator weaning and adjustment were performed at regular intervals (every 8 h) and as necessary, based on the general weaning guidelines and clinical practice of our respiratory therapy department [20]. 

### 2.4. Data Records in Our Previous Cohort

ICU admission date was considered as Day 0. The next date of ICU admission was defined as Day 1. The following patient data were recorded within 24 h after admission: age, sex, medical history, and APACHE II score. Adverse events were recorded within the first 3 days following admission. Arterial blood gases demonstrating the lowest PaO_2_/FiO_2_ ratio were used within 24 h after intubation with ventilator support. Driving pressure (∆P) was defined as the difference between Pplat and PEEP [21]. Driving pressures were recorded every 8 h per day. Serial mean data of ∆*P*_insp_, RR, *V*_T_, and PEEP were recorded daily for either 21 days or until the ventilator support was no longer required.

### 2.5. MP Calculation

MP for pressure-targeted ventilation was calculated every 8 h per day according to the simplified equation (Equation (1)) [14,22], using RR, tidal volume size (L) (*V*_T_), ∆*P*_insp_, and PEEP: (1)$$\mathrm{MP}\;\left(J/\min\right)=0.098\times \mathrm{RR}\times V_{T}\times \left(∆P_{\mathrm{insp}}+\mathrm{PEEP}\right)$$
where ∆*P*_insp_ is the change in airway pressure during inspiration. 

### 2.6. Statistical Analyses

Statistical analysis was performed using the Statistical Package for the Social Sciences (SPSS) version 17.0 for Windows (SPSS, Inc., Chicago, Illinois, USA). Differences in the continuous variables between the two groups were analyzed using the Student’s *t*-test. Differences in categorical variables between the ARDS and non-ARDS groups were compared using the Pearson chi-squared test or Fisher’s exact test. Analyses using the univariate binary logistic regression model were performed to study the association between the 28-day mortality and all variables. Statistically significant variables were entered into a multivariate binary logistic regression model to assess their independent contribution to the outcome. Since mechanical power is determined by RR, tidal volume, ∆*P*_insp_, and PEEP, ∆*P*_insp_ was not used as a covariable in multivariate analysis. The binary variables included in the model were coded as present or absent. The cutoff value used for the driving pressure and MP on Day 1 to predict 28-day mortality was identified according to the receiver operating characteristic (ROC) curve (Figure 1), with areas of 0.628 (*p* < 0.001) and 0.730 (*p* < 0.001) under the ROC curve, respectively. The area under the ROC curve of MP normalized to PBW (MP/PBW) was also calculated, which was 0.734 (*p* < 0.001). When the cutoff value for driving pressure was set at 19 cm H_2_O, the sensitivity and specificity were 58.1% and 57.7%, respectively. When the cutoff value for MP was set at 27 J/min, the sensitivity and specificity were 65.0% and 64.3%, respectively. A Kaplan–Meier graph was plotted to analyze the probability of death after ICU admission. Survival times of MP between MP < 27 J/min and MP ≥ 27 J/min were compared using a log-rank test. *P* values less than 0.05 were considered statistically significant. 

## 3. Results

A total of 493 patients were screened (Figure 2), out of whom 313 were enrolled in this study, while 180 patients were excluded. A total of 61 ARDS patients and 56 non-ARDS patients died within 28 days of ICU admission during the course of this study. Table 1 demonstrates the baseline clinical characteristics of patients with pneumonia in the ARDS and non-ARDS groups. Some patients with the PaO_2_/FiO_2_ ratio less than 300 mm Hg were classified as non-ARDS due to the absence of bilateral opacities on their chest radiography. The ARDS group had lower age and higher APACHE II score, MP, and driving pressure than the non-ARDS group. The incidences of shock, jaundice, and death were higher and mean age was lower in the ARDS group than those in the non-ARDS group. In total, pathogens were identified in 88.2% of the patients. The most frequently isolated pathogens, in decreasing order, were *Pseudomonas aeruginosa*, *Staphylococcus aureus, Acinetobacter baumannii*, *Klebsiella pneumoniae*, and *Escherichia coli*.

According to the binary logistic regression model, the variables that were independently associated with the 28-day mortality were the APACHE II score (odds ratio (OR), 1.044; 95% confidence interval (CI), 1.00–1.088), PaO_2_/FiO_2_ ratio (OR, 0.995; 95% CI, 0.993–0.997), shock (OR, 1.859; 95% CI, 1.009–3.425), thrombocytopenia (OR, 2.341; 95% CI, 1.287–4.261), and MP (OR, 1.048; 95% CI, 1.020–1.077) (Table 2). APACHE II score, shock, thrombocytopenia, and MP were positively correlated. 

The MP persisted at a high level in non-survivors and at a low level in survivors among the ARDS group, the non-ARDS group, and all patients (Figure 3). In all patients, MP in non-survivors was higher than in survivors during the 21 days recorded, except for Day 18. In the ARDS group, MP in non-survivors was higher than in survivors on Days 1–11, 16, and 20. In the non-ARDS group, MP in non-survivors was higher than in survivors on Days 1–14, and 17. The Kaplan–Meier curves showed the possibility of survival until 28 days after ICU admission for MP on Day 1 above and below 27 J/min among the ARDS group, the non-ARDS group, and all patients (Figure 4). Patients with MP < 27 J/min on Day 1 demonstrated significantly higher survival rates than those with MP ≥ 27 J/min in all patients and the non-ARDS group (*p* < 0.001). In the ARDS group, the patients with MP < 27 J/min did not show higher survival rates than those with MP ≥ 27 J/min (*p* = 0.067). 

## 4. Discussion

According to the multivariate regression analysis, MP on Day 1 and the PaO_2_/FiO_2_ ratio were independent factors associated with 28-day mortality. Patients with lower MP on Day 1 had better chances of survival compared to those with higher MP on Day 1. In the univariate regression analysis, driving pressure on Day 1 was a factor associated with 28-day mortality. In the ROC curves of the ARDS group, driving pressure did not discriminate survivors from non-survivors, but driving pressure could discriminate between survivors and non-survivors in all patients and in the non-ARDS group. This suggests that ARDS confounds the association between driving pressure and 28-day mortality. 

Our study first found that ventilated patients with pneumonia with or without ARDS demonstrated higher serial MP in non-survivors than in survivors from Day 1 to Day 11 of ICU admission. The MP between survivors and non-survivors was only statistically similar on Day 18 of ICU admission in all patients. Our study not only confirmed that baseline MP was associated with mortality [10,11,12,13], but also found that the association persisted for 11 days. In the Kaplan–Meier curves of the ARDS group, the two curves were similar within the first 10 days and were later evidently separated. This implies that the benefit of a lower MP requires a period of time to develop. Furthermore, deep sedation significantly reduced MP in patients with moderate to severe ARDS, thereby reducing the occurrence of VILI [13]. All of the findings above suggest that MP might be the cause of VILI, resulting in an increased mortality rate. Certainly, this is reasonable since all mechanical factors in ventilation—tidal volume, driving pressure, flow, resistances, RR, and PEEP—are different components of a unique physical variable, which is the energy delivered into the lung. Up to now, no study has reported the results of mortality using MP as a guide for ventilator settings to manage critically ill patients. Further studies are required to elucidate whether MP is a predictor or a cause. 

In our study, MP was an independent factor associated with 28-day mortality in ventilated patients with severe pneumonia. In an experiment on mild ARDS of rats, even at low *V*_T_, high MP promoted VILI [23]. In a computational study, MP showed a strong correlation with the relative risk of death across all ranges of driving pressures and PEEP [24]. Moreover, the areas under the ROC of MP were higher than those of driving pressure in the ARDS group, the non-ARDS group, and all patients in our study. This implies that MP might be a better predictor of 28-day mortality than *V*_T_ or driving pressure in patients with severe pneumonia with or without ARDS. Zhang’s study found 0.747 and 0.751 of areas under the ROC curve for MP and MP normalized to PBW, respectively [11]. For easy use in routine clinical practice, the calculation of MP instead of MP normalized to PBW might be sufficient. 

The MP was first determined with the simplified formula suggested by Gattinoni et al. for VCV (Equation 2) [8]:(2)$${\mathrm{MP}}_{\mathrm{VCV}}\;\left(\frac{J}{\min}\right)=0.098\times \mathrm{RR}\times V_{T}\times \left(\mathrm{peak}\;\mathrm{pressure}-\frac{1}{2}\mathrm{driving}\;\mathrm{pressure}\right)$$

As the equation for the calculation of MP is based on the assumption that the VCV has a linear increase of airway pressure during inspiration, it is not suitable for calculating MP during PCV [25]. For PCV, two accurate equations have been proposed, but both require using some parameters (resistance, respiratory system compliance) that are not usually continuously quantified and displayed in the ventilator [22,26]. Finally, we used the simplified formula for PCV proposed by Becher et al. [22] since this equation had acceptable accuracy and was routinely recorded by our respiratory therapeutist. Our study demonstrated that this simplified formula was easy to use, and the MP calculated had an acceptable discrimination for 28-day mortality. 

The most important current recommendation to provide lung protective ventilation in ventilated patients with pneumonia is the low tidal volume [3]. However, despite the use of lung protective ventilation proposed by the ARDSnet protocol, overall ICU and hospital mortality among ARDS patients is still higher than 40% [27]. Low tidal volume ventilation did not show any estimated benefit in patients with ARDS. It seems that low tidal volume ventilation alone is insufficient for the protection of the lung. In 2015, Amato et al. reported the results of a retrospective analysis and concluded that driving pressure was better associated with 60-day mortality in patients with ARDS than tidal volume [5]. Following this, Guerin et al. demonstrated this viewpoint [28], and a meta-analysis from Neto et al. showed that an increase in driving pressure was associated with more postoperative pulmonary complications [29]. Our previous study further found that higher driving pressure was associated with 28-day mortality in patients with severe pneumonia without ARDS [7]. In this study, the results showed that MP on Day 1 was independently and positively associated with 28-day mortality and had a better predicted value than driving pressure either in patients with ARDS or without ARDS. Our results support the hypothesis that a marker with several important indices is better than that with one index. The predicted value of MP, which included tidal volume, inspiratory pressure, RR, and PEEP, was better than driving pressure alone. 

This study has three limitations. First, this study was a single-center trial without thousands of case numbers. More studies are required to confirm our results. Second, chest radiograph findings might have been misinterpreted considering the presence of unilateral infiltrates or opacities due to the limitations of traditional chest radiography. Computed tomography may be a preferred technique for the detection of lung injury. Considering that the results from the ARDS and non-ARDS groups were similar, this did not significantly influence the final conclusion. Third, the mean MP in this study population was 27.8 J/min on Day 1, although the median MP in all ARDSnet trials was more than 28 J/min [11]. This is still relatively high.

## 5. Conclusions

Our findings imply that MP may be an important factor associated with 28-day mortality in ventilated patients with severe pneumonia, both in the ARDS and non-ARDS groups. The MP, which included tidal volume, inspiratory pressure, RR, and PEEP, had a better predicted value for 28-day mortality than driving pressure. The simplified formula for PCV to calculate MP was useful in patients ventilated with the PCV mode. 

## Figures and Tables

**Figure 1 diagnostics-11-01862-f001:**
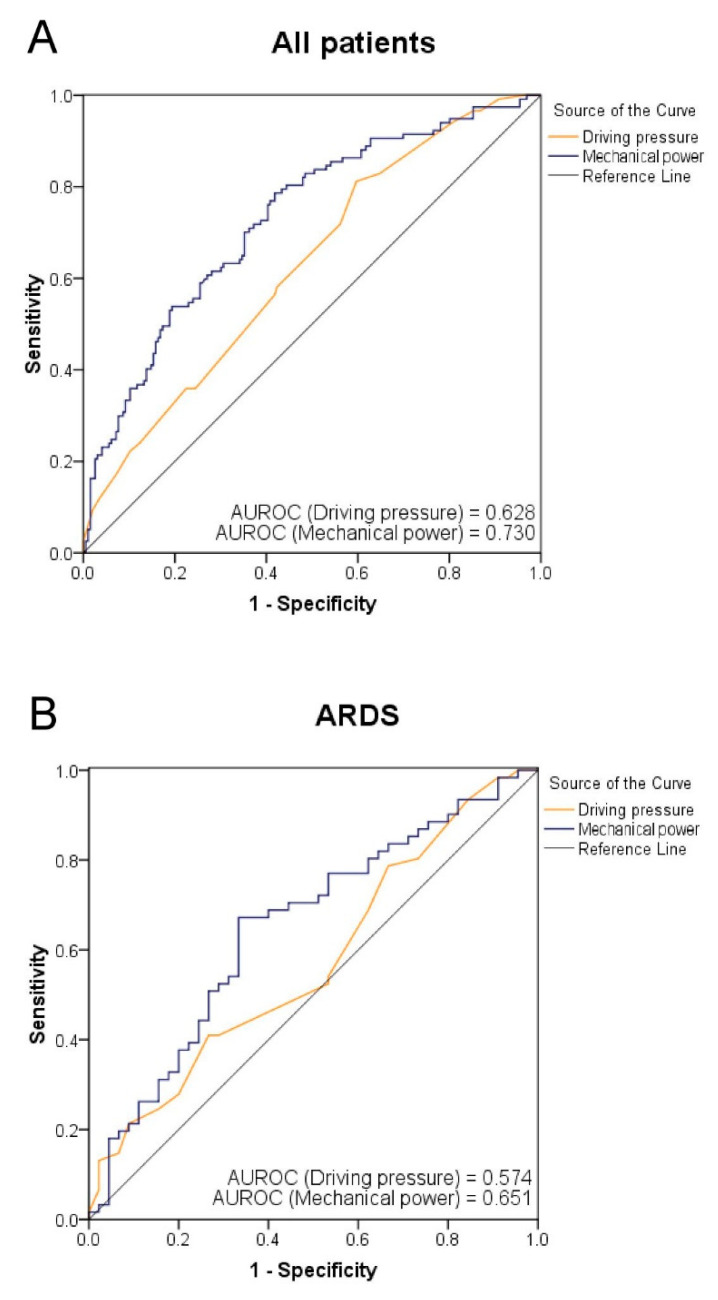
Receiver operating characteristic (ROC) curves of driving pressure and mechanical power on Day 1 for 28-day mortality among all patients (**A**), the acute respiratory distress syndrome (ARDS) group (**B**), and the non-ARDS group (**C**). The areas under the ROC (AUROC) were calculated. The AUROC for driving pressure and mechanical power were 0.628 (95% confidence interval (CI), 0.566–0.691; *p* < 0.001) and 0.730 (95% CI, 0.673–0.787; *p* < 0.001), respectively, in all patients. The AUROC for driving pressure and mechanical power were 0.574 (95% CI, 0.464–0.684; *p* = 0.197) and 0.651 (95% CI, 0.545–0.757; *p* = 0.008), respectively, in ARDS patients. The AUROC for driving pressure and mechanical power were 0.645 (95% CI, 0.565–0.725; *p* = 0.001) and 0.735 (95% CI, 0.655–0.814; *p* < 0.001), respectively, in non-ARDS patients.

**Figure 2 diagnostics-11-01862-f002:**
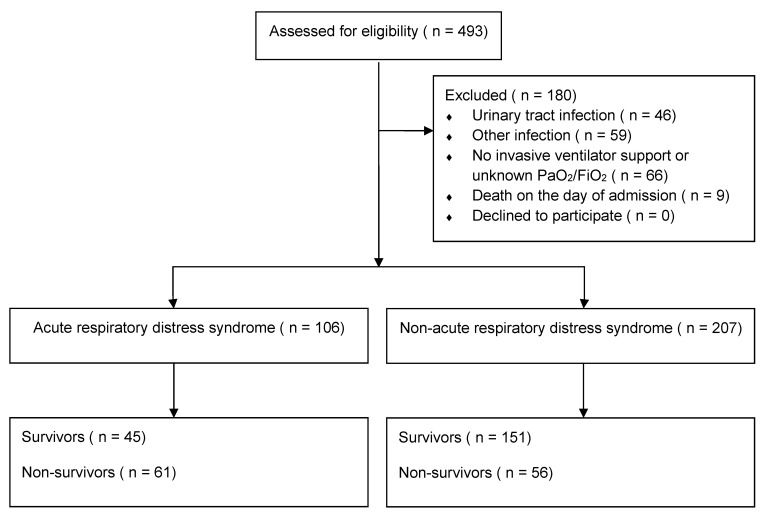
Flowchart of patient inclusion criteria. During the study period, we screened 493 patients who demonstrated symptoms of infection, and excluded 180 patients. The exclusion criteria include the following: diseases other than pneumonia, no invasive ventilation, an unknown ratio of arterial partial pressure of oxygen (PaO_2_) to fraction of inspired oxygen (FiO_2_), and death on the day of admission to the intensive care unit. In total, 313 patients were enrolled for analysis, and none of the patients withdrew from the study. Among them, 106 patients and 207 patients were classified as pneumonia with acute respiratory distress syndrome (ARDS) and non-ARDS, respectively. Sixty-one patients with ARDS and 56 patients with non-ARDS died during the course of the study.

**Figure 3 diagnostics-11-01862-f003:**
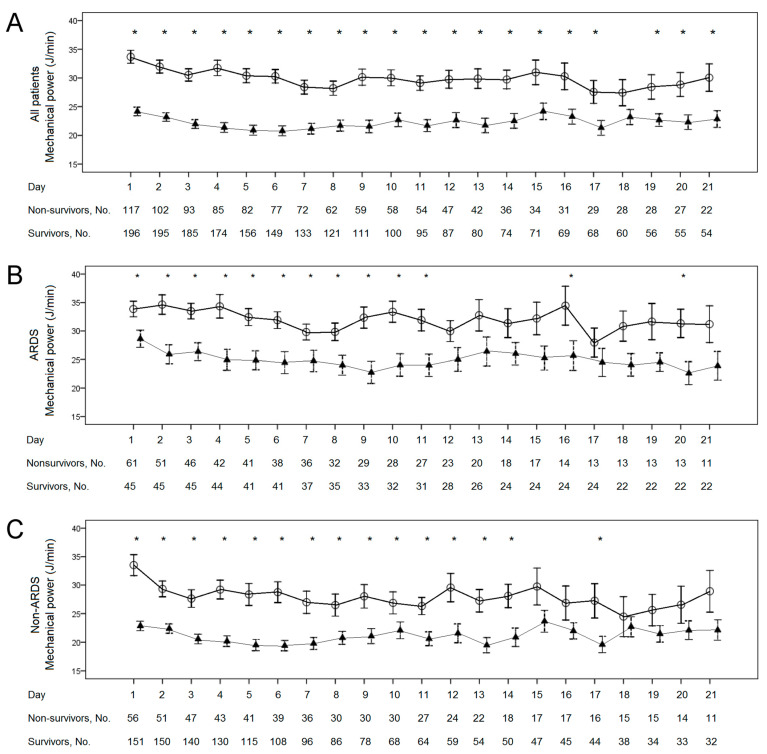
Error bars of serial mechanical powers (J/min, mean ± 1 standard error mean) from Day 1 to Day 21 between non-survivors and survivors. Asterisks represent significant differences between non-survivors and survivors using the Mann–Whitney U test. Hollow circle bars represent non-survivors, and solid triangle bars represent survivors. Serial mechanical powers were higher in non-survivors than in survivors among all patients during most of the 21-day period (**A**), the acute respiratory distress syndrome (ARDS) group (**B**), and the non-ARDS group (**C**).

**Figure 4 diagnostics-11-01862-f004:**
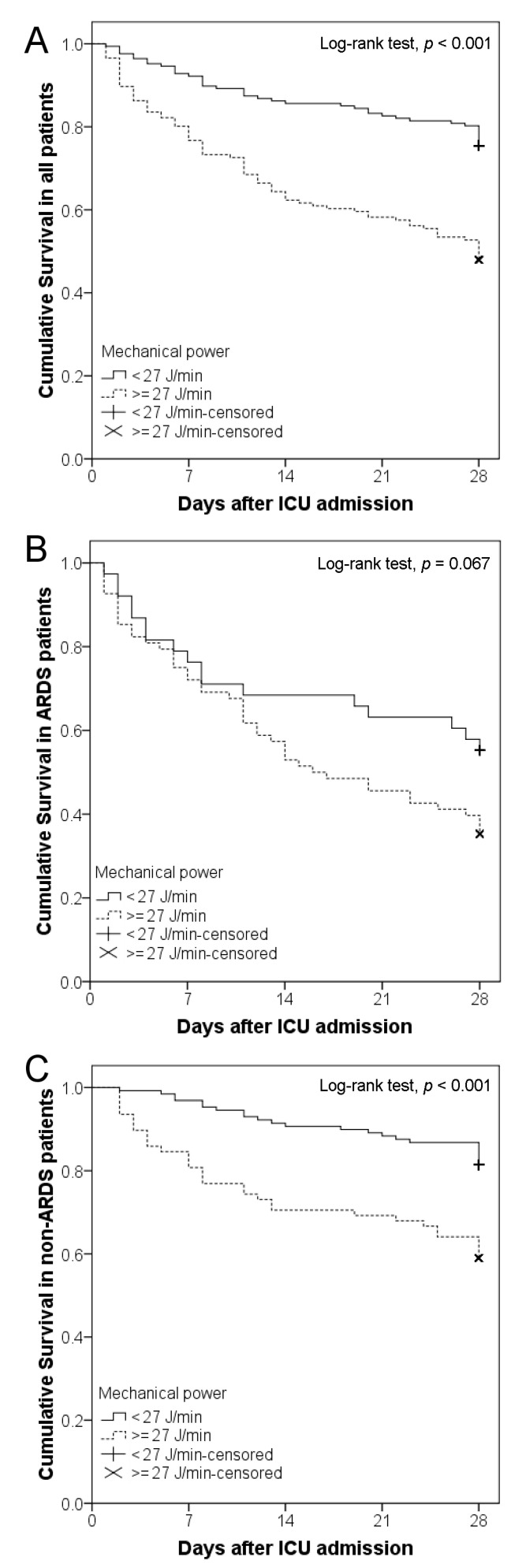
Kaplan–Meier graphs of 28-day intensive care unit (ICU) cumulative survival among all patients (**A**), the acute respiratory distress syndrome (ARDS) group (**B**), and the non-ARDS group (**C**) according to mechanical power on Day 1 of ICU admission (≥ 27, < 27 J/min).

**Table 1 diagnostics-11-01862-t001:** Clinical characteristics and outcomes of ventilated patients with pneumonia, according to subgroups.

Characteristics	ARDS (*n* = 106)	Non-ARDS (*n* = 207)	All Patients (*n* = 313)
Age, years *	69.1 ± 16.3	75.1 ± 13.1 ^†^	73.1 ± 14.5
APACHE II score *	27.6 ± 8.7	25.3 ± 6.8 ^†^	26.1 ± 7.6
Sex, No. (%)			
Male	75 (70.8)	137 (66.2)	101 (67.7)
Female	31 (29.2)	70 (33.8)	212 (32.3)
History, No. (%)			
COPD	24 (22.6)	49 (23.7)	73 (23.3)
CHF	7 (6.6)	20 (9.7)	27 (8.6)
Hypertension	40 (37.7)	97 (46.9)	137 (43.8)
Liver cirrhosis	10 (9.4)	13 (6.3)	23 (7.3)
Hemodialysis	6 (5.7)	21 (10.1)	27 (8.6)
Diabetes mellitus	19 (17.9)	68 (32.9) ^†^	87 (27.8)
Lung cancer	9 (8.5)	10 (4.8)	19 (6.1)
Non-lung cancer	29 (27.3)	34 (16.4)	63 (20.1)
Systemic lupus erythematosus	1 (0.9)	1 (0.5)	2 (0.6)
PaO_2_/FiO_2_ ratio (mm Hg) *	133.9 ± 70.3	350.6 ± 199.1 ^†^	277.2 ± 195.9
Adverse events, No. (%)			
Shock	59 (55.7)	68 (32.9) ^b^	127 (40.6)
Stage 2 or 3 acute kidney injury	47 (44.3)	78 (37.7)	125 (39.9)
GI bleeding	18 (17.0)	27 (13.0)	45 (14.4)
Thrombocytopenia	45 (42.5)	65 (31.4)	110 (35.1)
Jaundice	15 (14.2)	9 (4.3) ^†^	24 (7.7)
Pathogens, No. (%)			
*Pseudomonas aeruginosa*	16 (15.1)	45 (21.7)	61 (19.5)
*Staphylococcus aureus*	16 (15.1)	40 (19.3)	56 (17.9)
*Acinetobacter baumannii*	19 (17.9)	27 (13.0)	46 (14.7)
*Klebsiella pneumoniae*	19 (17.9)	20 (9.7)	39 (12.5)
*Escherichia coli*	7 (6.6)	17 (8.2)	24 (7.7)
*Streptococcus pneumoniae*	5 (4.7)	7 (3.4)	12 (3.8)
*Stenotrophomonas maltophilia*	4 (3.8)	6 (2.9)	10 (3.2)
*Mycobacterium tuberculosis*	2 (1.9)	7 (3.4)	9 (2.9)
*Candida species*	4 (3.8)	2 (1.0)	6 (1.9)
*Enterobacter species*	2 (1.9)	3 (1.4)	5 (1.6)
*Citrobacter species*	1 (0.9)	2 (1.0)	3 (0.9)
*Fusobacterium nucleatum*	1 (0.9)	1 (0.5)	2 (0.6)
*Corynebacterium jeikeium*	1 (0.9)	1 (0.5)	2 (0.6)
*Cryptococcus neoformans*	0 (0.0%)	1 (0.5)	1 (0.3)
*Staphylococcus epidermidis*	1 (0.9)	0 (0.0)	1 (0.3)
Mechanical power (J/min) on Day 1 *	31.7 ± 10.7	25.8 ± 12.2 ^†^	27.8 ± 12.0
Driving pressure (cm H_2_O) on Day 1 *	19.6 ± 4.5	18.1 ± 4.5 ^†^	18.6 ± 4.6
Death within 28 days, No. (%)	61 (57.5)	56 (27.1) ^†^	117 (37.4)

Abbreviations: ARDS = acute respiratory distress syndrome; APACHE = Acute Physiology and Chronic Health Evaluation; COPD = chronic obstructive pulmonary disease; CHF = congestive heart failure; PaO_2_ = arterial partial pressure of oxygen; FiO_2_ = fraction of inspired oxygen; GI = gastrointestinal. * Data are shown as mean ± standard deviation. ^†^ *p* < 0.05 compared with the ARDS group using the Mann–Whitney U test or the chi-squared test.

**Table 2 diagnostics-11-01862-t002:** Binary logistic regression to analyze the independent factors of 28-day mortality.

Variables	Univariate OR (95% CI)	*p* Value	Multivariate OR (95% CI)	*p* Value
Age	0.985 (0.970–1.001)	0.058		
APACHE II score	1.091 (1.055–1.128)	<0.001	1.044 (1.001–1.088)	0.045
Male	0.985 (0.604–1.607)	0.951		
COPD	0.906 (0.525–1.563)	0.722		
Congestive heart failure	0.824 (0.358–1.900)	0.650		
Hypertension	0.530 (0.330–0.851)	0.009	0.616 (0.343–1.107)	0.105
Liver cirrhosis	2.824 (1.181–6.749)	0.020	1.472 (0.472–4.597)	0.505
Hemodialysis	1.913 (0.866–4.227)	0.109		
Diabetes mellitus	0.587 (0.344–1.003)	0.051		
PaO_2_/FiO_2_ ratio (mm Hg)	0.004 (0.993–0.996)	<0.001	0.995 (0.993–0.997)	<0.001
Shock	3.859 (2.380–6.257)	<0.001	1.859 (1.009–3.425)	0.047
Stage 2 or 3 acute kidney injury	2.522 (1.573–4.042)	<0.001	1.411 (0.775–2.570)	0.260
Gastrointestinal bleeding	2.151 (1.137–4.068)	0.019	1.606 (0.669–3.856)	0.289
Thrombocytopenia	4.194 (2.560–6.872)	<0.001	2.341 (1.287–4.261)	0.005
Jaundice	3.056 (1.292–7.227)	0.011	0.766 (0.249–2.354)	0.641
Mechanical power (J/min) on Day 1	1.076 (1.051–1.101)	<0.001	1.048 (1.020–1.077)	0.001
Driving pressure (cm H_2_O) on Day 1	1.126 (1.065–1.190)	<0.001		

Abbreviations: OR = odds ratio; CI = confidence interval; APACHE = Acute Physiology and Chronic Health Evaluation; COPD = chronic obstructive pulmonary disease; ARDS = acute respiratory distress syndrome.

## Data Availability

The datasets generated for this study are available upon request from the corresponding author.

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
