# Peer review of "The Association between Mechanical Power and Mortality in Patients with Pneumonia Using Pressure-Targeted Ventilation"

_diagnostics, 2021, doi:10.3390/diagnostics11101862_

Round 1

Reviewer 1 Report

Thank you very much for giving me the opportunity to review the manuscript entitled “The association between mechanical power and mortality in pressure-targeted ventilated patients with pneumonia”. The authors investigated the association between mechanical power (MP) and mortality in patients with pneumonia, and demonstrated MP is associated with mortality both ARDS and non-ARDS patients with pneumonia. The findings are very interesting. However, there are some critical concerns as follows.

  1. Why did the author use MP on the first day as a representative value of MP during mechanical ventilation? Does the time-weighted average of MP more accurately represent the true stress to the lungs?
  2. High MP observed in the dead patients might be the results of low oxygenation and high carbon dioxide levels in these patients. Therefore, baseline respiratory status must be adjusted in the multivariate analysis.
  3. The authors included variables which are significant in the univariate analysis into the multivariate analysis. However, this method is not appropriate in this study. The mechanical power is determined by respiratory rate, tidal volume, driving pressure and PEEP. How are the contributions of each of these variables for mortality? It is difficult to interpret the statistical model including both MP and driving pressure as covariates. Please provide more convincing evidence demonstrating the association of these variables. Utilization of mediation analysis, which is used in the Amato’s NEJM paper published in 2015, may be a solution for these problems.
  4. Please perform statistical analysis using the propensity scores predicting the probability of receiving MP more than 27J/min.

Reviewer 2 Report

Many thanks for the opportunity to Review the Manuscript entitled: The association between mechanical power and mortality in pressure-targeted ventilated patients with pneumonia.

Authors explored data of 313 patients with severe pneumonia dividing into ARDS-group and not ARDS-group and analyzing the clinical demographic characteristics of these patients and investigated the association between 28-day mortality and MP (mechanical power) in patients with severe pneumonia.

However,  authors should define in the method in the disease definition the diagnosis of pneumonia because chest –X ray it is not sufficient alone to make the diagnosis. Therefore, also in the conclusion authors reported as a study limitation that chest radiograph findings might have been misinterpreted considering the presence of unilateral infiltrates or opacities due to the limitations of traditional chest radiography and they didn’t use chest CT that also is not sufficient without the laboratory confirmation.

Authors should specify the diagnostic criteria for pneumonia with also laboratory examinations (blood cultures, analysis on pleural effusion, BAL) and should specify in the results also creating a different table the percentages of the pathogens causing the pneumonia (Pseudomonas ..(how many?)..Klebsiella..Aspergilus..viral.. Cytomegalovirus?), Authors should also specify in the Table 1 how many in the ARDS and not ARDS group were in an immunodepressive state and found also if there is a correlation also with MP. Authors should specify in the results the most frequent pathogens causing pneumonia in the ARDS and not ARDS group.

I don’t find the Fig 4 in the text

I suggest to add also some chest-X ray images of the Ards and not-Ards group

Round 2

Reviewer 1 Report

The manuscript has been significantly improved. I have no further concerns.